

# Technical note: Geodynamic Thermochronology (GDTchron) – A Python package to calculate low-temperature thermochronometric ages from geodynamic numerical models

Dylan A. Vasey[1], Peter M. Scully[1], John B. Naliboff[2], Sascha Brune[3,4]

[1]Department of Earth and Climate Sciences, Tufts University, Medford, MA, 02155, USA
[2]Department of Earth and Environmental Science, New Mexico Institute of Mining and Technology, Socorro, NM, 87801, USA
[3]GFZ Helmholtz Centre for Geosciences, Telegrafenburg, 14473 Potsdam, Germany
[4]Institute of Geosciences, University of Potsdam, 14476 Potsdam-Golm, Germany

*Correspondence to*: Dylan A. Vasey (dylan.vasey@tufts.edu)

**Abstract.**

Low-temperature thermochronology provides a powerful means of extracting quantitative information on the thermal evolution of different tectonic settings from rocks exposed at the surface of the Earth. Geodynamic numerical models enable tracking the entire thermal structure of simulated tectonic settings throughout their evolution. Despite the highly complementary nature of these two approaches, few geodynamic modeling studies have used the thermal information in models to predict thermochronometric ages as a means of comparing model results with observational data. Here, we present Geodynamic Thermochronology (GDTchron): an open-source Python package designed to forward model large numbers of low-temperature thermochronometric ages from time-temperature paths output by geodynamic numerical models. This package uses existing techniques to estimate apatite (U-Th)/He, apatite fission track, and zircon (U-Th)/He ages from time-temperature paths in a parallelized workflow that enables faster computation on multicore processors and high-performance computing systems. It is designed to extract the temperature of many selected particles over multiple timesteps. Our workflow is built on typical output files from geodynamic models containing particle location, time, and temperature, and we use an interpolation scheme to allow new particles to inherit the thermal histories of their nearest neighbors. GDTchron can be applied to any tectonic setting, though for results to be comparable to nature, geodynamic models should account for erosion and sedimentation. We demonstrate the functionality of this software with a highly simplified geodynamic model of uplift and a more complicated model of rift-inversion orogenesis with the aim of encouraging community participation in broadening future development.

## 1 Introduction

Interpretation of low-temperature thermochronometric data and ages has become increasingly sophisticated in recent years, enabling extraction of more detailed information on the thermal history of many tectonic settings from the (U-Th)/He and





fission track systems (e.g., van der Beek and Schildgen, 2023; Enkelmann and Garver, 2016; Flowers and Peak, 2025; Fosdick et al., 2024; Gallagher and Parra, 2020; Guenthner, 2021; Ketcham, 2024). Simultaneously, advances in geodynamic numerical modeling have enabled increasingly high-resolution simulations of these tectonic settings with increasingly realistic approximations of a range of geologic processes (e.g., Balázs et al., 2021; Fraters and Billen, 2021; Glerum et al.,

2024; Jourdon and May, 2022; Neuharth et al., 2022; Wolf et al., 2021; Zwaan et al., 2025). However, the non-uniqueness of the tectonic processes and thermal histories that produce thermochronometric data and the difficulties of validating geodynamic models with observational data remain major challenges. Increased interaction between these fields has the potential to improve the interpretive capabilities of each. Geodynamic models track temperature over time with complete knowledge of the processes that impact the temperatures recorded, while thermochronometric data at the surface of the Earth

are quantitative constraints against which geodynamic models could be validated. Although thermokinematic models have often been used to predict low-temperature thermochronometric ages (e.g., Braun, 2003; Braun et al., 2012; Capaldi et al., 2022; Curry et al., 2021; Erdős et al., 2014), only a few studies have used the outputs of geodynamic models to predict such ages for comparison with natural systems (Jourdon et al., 2018; Ternois et al., 2021).

Here, we present Geodynamic Thermochronology (GDTchron): a Python package designed to facilitate the forward modeling of large numbers of low-temperature thermochronometric ages using time-temperature (t-T) histories extracted from geodynamic numerical models (Vasey and Scully, 2025). Our package employs existing diffusion and annealing models implemented in the widely-used HeFTy inverse modeling program (Ketcham et al., 1999, 2000, 2011; Ketcham, 2005) to estimate apatite (U-Th)/He (AHe), apatite fission track (AFT), and zircon (U-Th)/He (ZHe) ages from prescribed t-

T paths in a parallelized workflow that can take advantage of multiprocessing and high-performance computing (HPC). The package contains tools for extracting t-T paths for unique particles from commonly used geodynamic model output file formats across multiple timesteps, and an interpolation scheme allows particles created during a model run to inherit thermal information from their nearest neighbors. We present two example geodynamic models illustrating potential use-cases for this package: 1) a highly simplified model of uplift that illustrates the variable behavior of the AHe, AFT, and ZHe systems

and 2) a more complex model of rift-inversion orogenesis where geodynamic processes are coupled with erosion and sedimentation routines, with results containing thermal information about both rift-related and orogenic exhumation. We note that this is an early version of software that we envision as a community-developed tool, and we outline current limitations and paths forward for future development.

## 2 Forward modeling of (U-Th)/He and apatite fission track ages

At the core of GDTchron are modules designed to forward model low-temperature thermochronometric ages for individual mineral grains given a time-temperature history. Although these models are relatively simple, they provide a first order, computationally efficient age appropriate for assessing the patterns of ages within geodynamic models. Because they are





open-source and contained within a publicly hosted Python package, they can be easily extended by the community for more complex models and used for education on the fundamentals of low-temperature thermochronology. Note that, because uncertainties reported for observed low-temperature thermochronometric ages are typically analytical uncertainties, our model ages based on predefined thermal histories are reported without uncertainties.

## 2.1 Apatite and zircon (U-Th)/He

We model the (U-Th)/He system by combining ingrowth of He into a mineral crystal via decay of $^{238}$U, $^{235}$U, and $^{232}$Th with temperature-controlled diffusion of He out of the crystal lattice. We use the finite difference solution to the diffusion equation outlined by Ketcham (2005) to approximate He diffusion in a sphere. This approach divides the radius of the sphere into a series of nodes, with loss of He at each node a function of temperature and distance from the edge of the sphere. The concentration of He at each node is then scaled to the relative volume of a spherical shell corresponding to the position of the node along the radius. The He concentrations in each shell are summed to obtain the total He within the sphere, which is then used to calculate a (U-Th)/He age. The effect of α ejection on He concentration and the resulting age is approximated by reducing the amounts of U and Th used to model He ingrowth by a factor determined by the distance from the edge of the sphere and the α stopping distance for the mineral, after Ketcham et al. (2011). The diffusivity of He within a particular mineral system is modeled using empirically-derived frequency factors ($D_0$) and activation energies ($E_a$) reported for apatite (Farley, 2000) and zircon (Reiners et al., 2004).

## 2.2 Apatite fission track

The apatite fission track (AFT) system is modeled by simulating annealing of fission tracks within an apatite crystal at a given temperature for each timestep within a time-temperature path, following the approach outlined in Ketcham (2005). Annealing is approximated using the fanning curvilinear model of Ketcham et al. (1999), which relates normalized c-axis projected fission track lengths to time and temperature via a set of six fitted parameters and results in in slightly curved lines of constant annealing that fan out from a single point on an Arrhenius plot. For each timestep, new normalized c-axis projected lengths are calculated based on the average temperature of the timestep, the length of the timestep, and the equivalent time needed to produce the initial normalized c-axis projected lengths prior to the timestep at the temperature of the timestep. The c-axis projected lengths are adjusted based on an input etch pit diameter ($D_{par}$) that approximates chemical variations within apatite crystals and converted to normalized fission track densities (Ketcham et al., 2000, 1999; Ketcham, 2005). The densities are summed to calculate a total AFT age, and the normalized c-axis projected lengths are converted to mean c-axis projected lengths and combined in order to visualize the shape of the length distributions.

## 2.3 Example forward models from a time-temperature path

Figure 1 shows a prescribed time-temperature path for a sample that resides at 100°C from 35 Ma to 15 Ma, cools at a rate of 25°C/Myr from 15 Ma to 11 Ma, and then remains at 0°C from 11 Ma. For the apatite (U/Th)/He (AHe) system, this results




in an age of 13.9 Ma, which represents the rapid cooling of the sample through the entire AHe partial retention zone between

15 Ma and 11 Ma. For the zircon (U-Th)/He (ZHe) system, this path results in an age of 34.8 Ma, corresponding to nearly

the full duration of the 35 Myr history, given that the sample remains at a temperature largely below the ZHe partial

retention zone throughout this history. The apatite fission track (AFT) age is 21.3 Ma, an intermediate value between the 35

Ma start of the path and the 15 Ma to 11 Ma cooling period, indicating that the sample stayed within the AFT partial

annealing zone until 15 Ma and then cooled below the partial annealing zone.

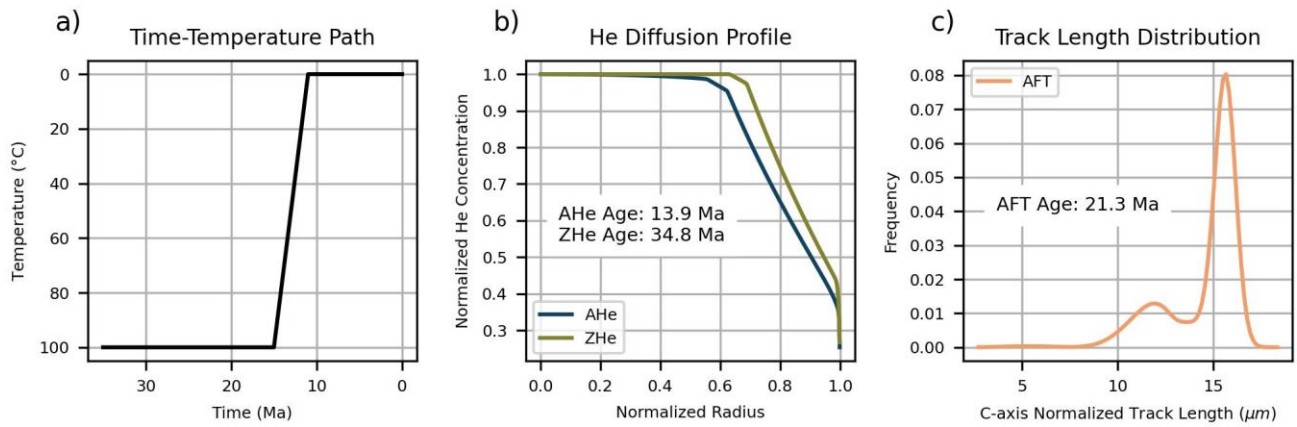

**Figure 1:** Example forward modeling results for a single time-temperature path. **a)** Time-temperature path over 35 Myr with cooling from 100°C to 0°C from 15 Ma to 11 Ma. **b)** He diffusion profiles and ages after 35 Myr for apatite (AHe) and zircon (ZHe) grains with 100 ppm each of U and Th and a radius of 50 μm. **c)** C-axis normalized track length distributions and age for apatite (AFT) grain with $D_{par}$ of 1.75 after 35 Myr.

## 3 Scaling for large numbers of time-temperature paths

The primary new functionality in GDTchron is the ability to efficiently use the large number (thousands to millions) of time-temperature (t-T) paths output by geodynamic models to predict synthetic AHe, AFT, and ZHe ages throughout the model domain and at each timestep of the model run.

### 3.1 Parallelized forward modeling

To make use of multi-core processors available on high-performance computing (HPC) nodes and personal computers, we

parallelize our forward models using the Python package Joblib. This allows the processor to calculate multiple t-T paths

simultaneously, depending on the number of cores available, in order to speed up the calculation of many t-T paths. Within

this parallelized framework, the choice of batch size can be particularly important for maximizing model efficiency. Batch

size refers to the number of jobs (e.g., t-T paths) sent to each worker (i.e., core) at once; higher batch size decreases the

number of times jobs need to be routed to the worker while increasing the amount of computation routed each time. We find

that optimal batch size varies significantly depending on the length of the t-T path, the number of total paths, and the device



used. For geodynamic model outputs, where the number of paths is usually greater than 10,000 and the individual calculations at each timestep are relatively cheap, batch sizes of ~100 to 1,000 generally lead to the most efficient parallelization.


In an ideally parallelized workflow, doubling the number of cores will halve the computation time, though the overhead required to set up parallelization, as well as the increased memory usage of parallelized workflows, will dampen this effect. Figure 2 shows scaling results up to 64 cores on a node on the Tufts HPC cluster for prediction of 96,000 AHe ages for t-T paths with random temperatures between 0°C and 75°C assigned every 100,000 years over 40 Myr. Scaling results are

generally good, with the computation taking 1,534 seconds on 1 core and 37 seconds on 64 cores (Figure 2). The resulting ages are generally between 10 Ma and 22 Ma, reflecting t-T paths partially in and partially below the AHe partial retention zone (e.g., Reiners and Brandon, 2006).

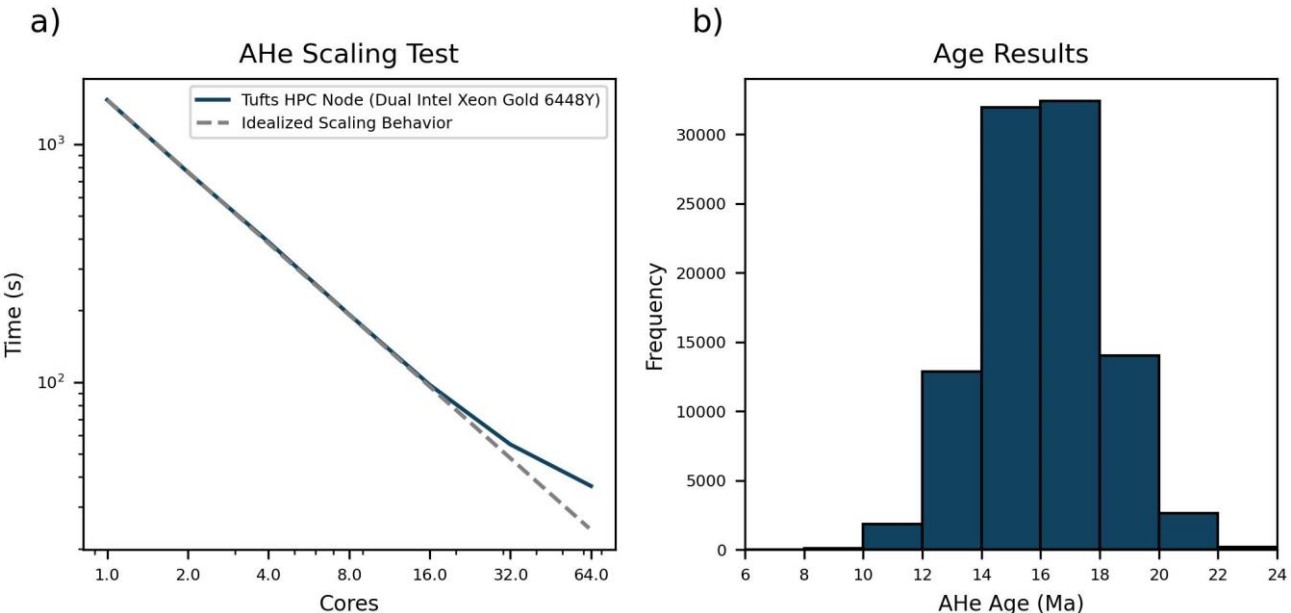

**Figure 2: a)** Scaling results for Tufts HPC node producing apatite (U-Th)/He (AHe) ages from 96,000 synthetic time-temperature (t-T)
paths with pseudorandom temperatures between 0°C and 75°C assigned every 100,000 years over 40 Myr. **b)** Histogram of resulting AHe ages produced from the 96,000 t-T paths.

### 3.2 Processing outputs from geodynamic modeling software

Geodynamic models typically output results at evenly spaced timesteps, and such results are frequently embedded within complex data structures that can accommodate the irregular geometries and many parameters tracked in such models.
GDTchron is currently designed to work with file formats from the Visualization Toolkit (VTK) software, which is commonly used for visualization of three-dimensional scientific data. These file formats are the standard output of the open-





source geodynamic modeling software ASPECT (Advanced Solver for Problems in Earth's Convection and Tectonics; Bangerth et al., 2024; Heister et al., 2017; Kronbichler et al., 2012), as well as other geodynamic codes (e.g., LaMEM; Kaus et al., 2016; pTatin3D; May et al., 2014; ELEFANT; Thieulot, 2014). For use with ASPECT, GDTchron requires that
models track material properties using particle-in-cell (PIC) methods (e.g., Gassmöller et al., 2018, 2019), since this allows the temperature of the same particle to be tracked across multiple timesteps.

GDTchron uses the Python-based visualization package PyVista (Sullivan and Kaszynski, 2019) to extract all particle identification numbers and their corresponding temperatures from the output VTK file of each ASPECT timestep. The
temperatures and time elapsed between timesteps are distributed to the forward models described in Section 2 to update the He profile or fission track length distribution for that particle and to calculate a thermochronometric age at that timestep. The ages are added as a new scalar field to the imported VTK file. The He profile or length distribution for each particle is retained and then used as the starting point for the same particle at the subsequent timestep, preventing duplicate computation and allowing ages to be calculated at all timesteps for all particles that are present throughout the model run.


In PIC geodynamic models, individual particles are added and deleted as necessary to achieve optimal particle density in each cell during the model run. This results in the frequent addition of particles with no prior thermal history. We address this problem at a first-order level with an interpolation scheme using $k$-d trees (Bentley, 1975; Virtanen et al., 2020), in which new particles are assigned the He profile or track length distribution of the particle with the needed information that
they are closest to in terms of physical difference (Figure 3). We note that using such interpolation both increases computation time and introduces the possibility of assigning a particle a thermal history that it would not have experienced, particularly in the case of particles that are added near the surface of a geodynamic model to approximate sedimentation, as discussed in Section 5. As a result, this is an optional component of the software that can be turned off such that only particles present at all timesteps are assigned thermochronometric ages.





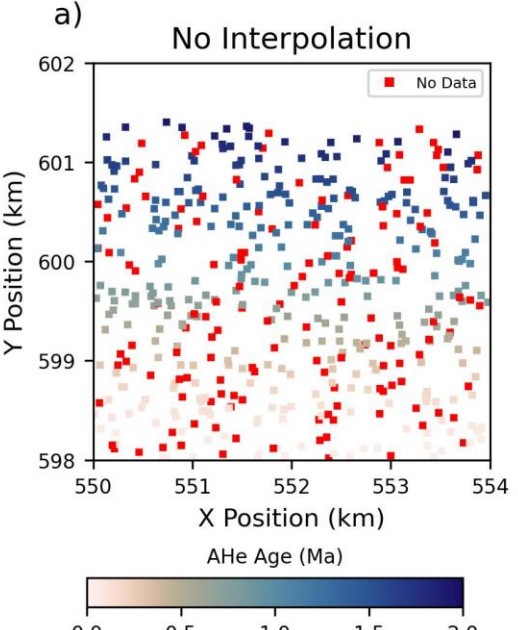
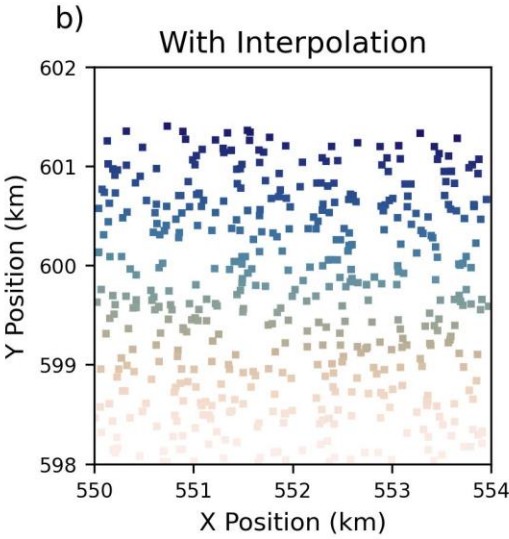


**Figure 3:** Individual particles with assigned apatite (U-Th)/He (AHe) ages after 36 Myr of a rift inversion model. **a)** With no interpolation of thermal history, particles added during the model run have no result (red). **b)** With interpolation as described in the text, all particles are assigned a thermal history and AHe age. Colormaps here and below from Crameri et al. (2020).

### 3.3 Application to a simple uplift model

To illustrate the above functionality, we designed a simple PIC ASPECT model consisting of a two-dimensional box 100 km wide and 20 km deep (Figure 4). This box is assigned a linear geothermal gradient of 30°C with a surface temperature of 0°C. For 10 Myr, the box is kept static, such that the shallowest particles in or above the partial retention zones for AHe and ZHe and the partial annealing zone for AFT would accumulate He and fission tracks. At 10 Myr, the bottom of the box is pushed upwards at a rate of 1 mm/yr, allowing material to flow out the top of the box while the temperature structure of the

box remains constant. This is designed to simulate 5 km of uplift with perfectly efficient erosion maintaining the original surface of the model. This uplift continues until 15 Myr, at which point the model remains static again until 20 Myr.





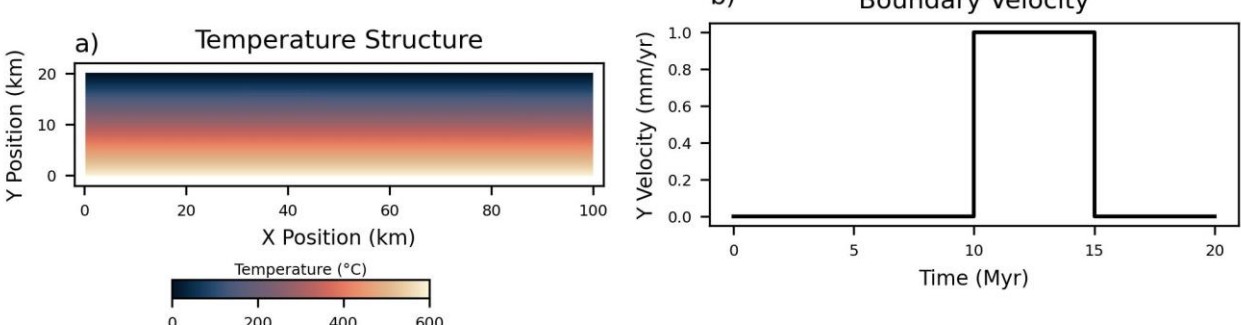

**Figure 4:** Initial conditions for simple model of uplift of a 2D box. **a)** Model domain consisting of a 100 x 20 km box with a surface temperature of 0°C and a linear geothermal gradient of 30°C. **b)** Velocity imposed on the bottom of the model during the model run over
20 Myr

Figure 5 shows the resulting AHe, AFT, and ZHe ages after processing the output of this model, which consists of ~100,000 particles distributed over 200 timesteps of 100,000 years, with GDTchron. For each of these systems, during the first 10 Myr, the shallowest particles begin with ages of 0 Ma and yield progressively older ages with time, with the ages at the surface of the model essentially equal to the model runtime. Ages young with depth until reaching 0 Ma below the partial
retention/annealing zones of each respective system, where diffusion/annealing outpace He/fission track production. During uplift from 10 Myr to 15 Myr, the oldest ages at the surface are removed while material with younger ages is brought to the surface, smoothing the transition from young ages at depth to old ages at the surface. Surface AHe ages are the youngest, whereas ZHe ages are the oldest, reflecting the erosion of nearly the entire original AHe partial retention zone while much of the original ZHe partial zone remains intact. During the final 5 Myr, material near the surface gets progressively older again,
resulting in final surface ages and age distributions at depth that are the cumulative effect of quiescence, uplift, and additional quiescence.



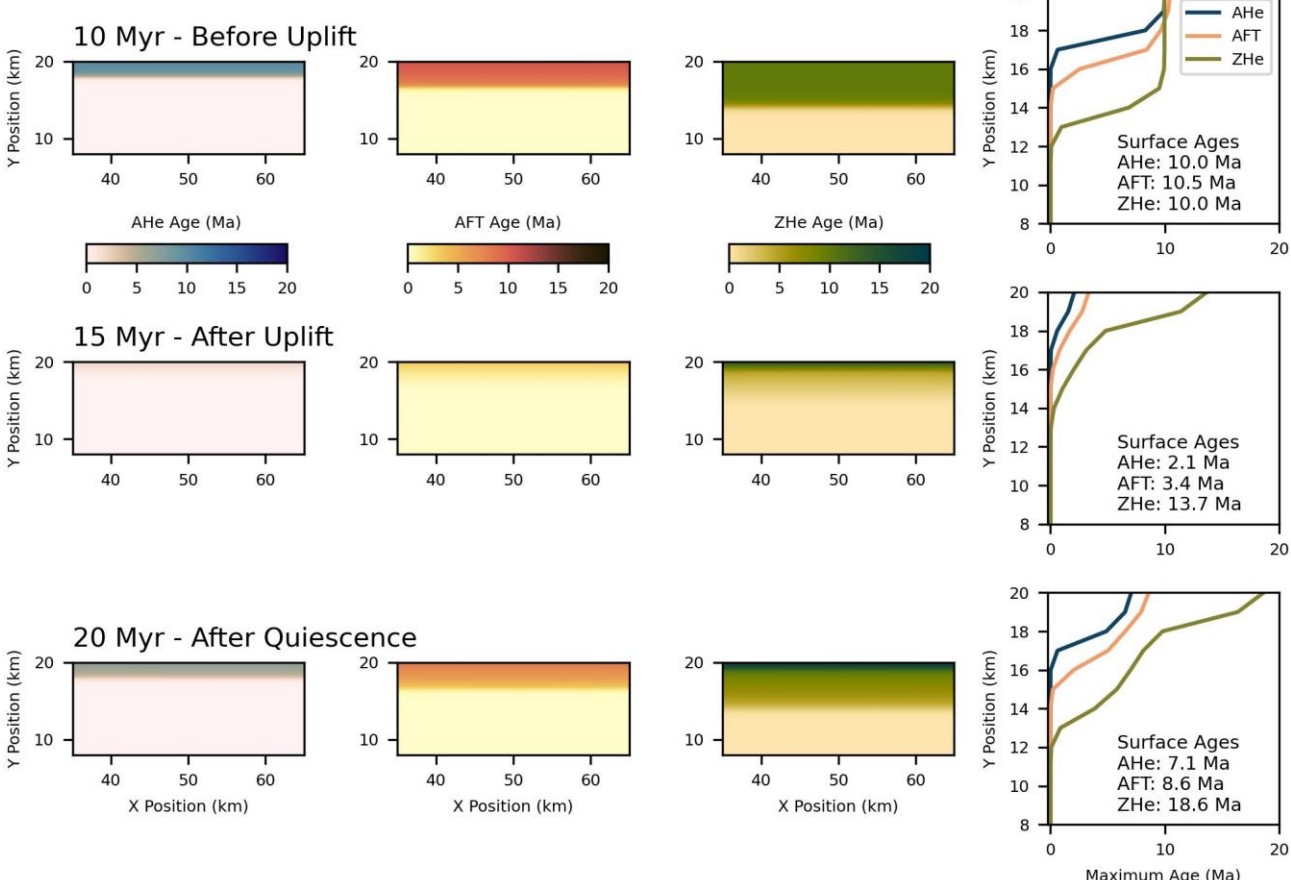

**Figure 5:** Results from simple uplift model for the apatite (U-Th)/He (AHe), apatite fission track (AFT), and zircon (U-Th)/He systems at key moments in the model run. Left three columns show distribution of ages for each of the three systems; right column shows age as a function of depth in the model domain, with age at the surface of the model reported. Top row (10 Myr) shows results after 10 Myr of ingrowth of He and fission tracks with no uplift. Center row (15 Myr) shows results after 5 Myr of uplift at 1 mm/yr, and bottom row (20 Myr) shows results after an additional 5 Myr of quiescence following uplift.

This example model demonstrates the potential utility of our software as a means of assessing patterns of low-temperature thermochronometric age distribution in a variety of geodynamic settings. One could explore the effects of varying fault network evolution, geothermal gradients, plate motions, and crustal rheology on expected age distribution and use real thermochronology data as a means of validating choices of model parameters.

## 4 Application to complex geodynamic modeling of rift-inversion orogenesis

The uplifting box model described above is deliberately simplistic in order to demonstrate expected patterns of thermochronometric ages in a highly controlled system. More complex geodynamic models can deploy realistic deformation,



geothermal gradients, surface processes, mantle convection, radiogenic heat production, shear heating, and other features that impact the thermal history material within the model. Since all these processes can impact the final ages derived from rocks at the surface of the Earth, these models provide a means of testing the sensitivity of ages to variations within these parameters and of using patterns of ages from natural settings to validate geodynamic models.

To demonstrate one of many possible examples, Figure 6 shows the results of processing an ASPECT model of rift-inversion orogenesis with GDTchron to generate synthetic AHe, AFT, and ZHe ages. Rift-inversion orogens are compressional mountain belts in which a continental rift present prior to compression serves as a preexisting lithospheric weakness that localizes deformation (e.g., Cooper et al., 1989; Vasey et al., 2024; Zwaan et al., 2025). These orogens, which include the Pyrenees (e.g., Muñoz, 1992), High Atlas (e.g., Beauchamp et al., 1999), and Greater Caucasus (e.g., Vincent et al., 2016)

have multi-phase thermal histories that may include periods of extension, tectonic quiescence, and compression, all of which may impact the final thermochronometric ages recorded in rocks exposed within the present-day mountain belt. Thus, predicting synthetic ages within models of rift-inversion orogenesis serves as a means of deconvolving the impacts of the individual phases on the final thermal imprint recorded by thermochronology. Although thermokinematic models like Pecube (Braun, 2003; Braun et al., 2012) and Pecube-D (Ehlers, 2023) have been used to predict thermochronometric ages in

such orogens (e.g., Capaldi et al., 2022; Curry et al., 2021; Erdős et al., 2014), we are only aware of one prior study focused on the Pyrenees (Ternois et al., 2021) in which fully dynamic models of rift-inversion orogens were used to predict thermochronometric ages. This study extracted the thermal history of only 14 tracer particles, which were then manually passed to the software QTQt (Gallagher, 2012) for forward modeling.



**Figure 6:** Results from rift inversion model at the rift (left column, 16 Myr) and the inversion stage (right column, 36 Myr), designed after Model 1 in Vasey et al. (2024) with the addition of surface processes from the code Fastscape. Top row shows distribution of compositional layers, plastic strain, and temperature structure. Middle row shows distribution of apatite (U-Th)/He (AHe) ages for the same part of the model domain. Bottom row shows AHe, apatite fission track (AFT), and zircon (U-Th)/He ages at the surface along the X component of the model domain.

Our example model is based on Model 1 from Vasey et al. (2024) but is modified to provide more realistic surface processes through coupling of ASPECT with the surface processes code Fastscape (Braun and Willett, 2013; Neuharth et al., 2022; Yuan et al., 2019a, b). The models in Vasey et al. (2024) used hillslope diffusion to approximate surface processes (Sandiford et al., 2021), and we found that such diffusion dampened exhumation sufficiently to prevent exposure of material with young thermochronometric ages at the surface of the model. This 2D model simulates rift inversion by first extending a 1000 x 600 km block divided into upper crust, lower crust, mantle lithosphere, and asthenosphere for 16 Myr at 0.5 cm/yr to create a continental rift. At 16 Myr, the model then undergoes convergence at 1 cm/yr for 20 Myr to invert the rift and create



a mountain belt. These specific model parameters result in an asymmetric mountain belt, with the lithosphere on one side of the model underthrust beneath the lithosphere on the other side, and development of major shear zones approximating the orogenic wedge of a real mountain belt (e.g., Beaumont et al., 1996; Dahlen, 1984; Willett et al., 1993).


During rifting, the flanks of the rift are uplifted and exhumed relative to the down-dropping basin, resulting in slightly younger AHe (~10 Ma) and AFT (~14 Ma) ages at the surface of the rift flanks relative to elsewhere along the surface after 16 Myr. Insufficient material is exhumed to produce younger ZHe ages, given the deeper partial retention zone of this system (e.g., Reiners and Brandon, 2006). During rift inversion, rapid exhumation in the center of the model occurs due to

lithospheric thickening and uplift, leading to a zone of young (<5 Ma) AHe, AFT, and ZHe ages at the end of orogenesis. Notably, this zone widens as deformation propagates towards the forelands on either side of the doubly-vergent orogenic wedge, overprinting the rift flank signal on the right (prowedge) side of the model. On the left (retrowedge) side of the model, the effects of rifting are preserved as a zone of slightly younger AHe (~30 Ma) and AFT (~35 Ma) ages. Because our software allows us to see the evolution of these thermochronometric systems throughout the model run, it is possible to

clearly separate the effects of rifting and inversion. Modifying the parameters of rift inversion to change the structure of the rift and orogen will likely also change the patterns of thermochronometric ages, which can then be compared to the increasing quantity of observational data from rift-inversion orogens like the Pyrenees (e.g., Curry et al., 2021), High Atlas (e.g., Lanari et al., 2020), and Greater Caucasus (e.g., Vincent et al., 2020).

## 5 Limitations and future directions

We emphasize that the current version of our software represents only the first step towards more fruitful combination of geodynamic modeling with low-temperature thermochronology. Currently, the software only employs a simple He diffusion model for AHe (Farley, 2000) and ZHe (Reiners et al., 2004), whereas considerable work has gone into developing models that take radiation damage and annealing into account (e.g., Enkelmann and Garver, 2016; Flowers et al., 2009; Guenthner, 2021; Guenthner et al., 2013, 2014; Shuster et al., 2006). Likewise, only the model of Ketcham et al. (1999) for fission track

annealing in the AFT system is currently used, and numerous improvements in modeling annealing have been proposed since that time (e.g., Ketcham, 2019; Ketcham et al., 2007; Tamer and Ketcham, 2020), and models of the zircon fission track (ZFT) system could also be implemented (e.g., Yamada et al., 2007). Our current software also does not allow for U-Th zonation (e.g., Ault and Flowers, 2012; Farley et al., 2011; Hourigan et al., 2005) or geometries other than a sphere (e.g., Glotzbach et al., 2019; Herman et al., 2007) when modeling He diffusion. Implementing these alternative kinetic models in

future versions would allow one to see if and how these choices would affect the large-scale distribution of ages within a particular geodynamic setting.



Importantly, the ages produced when applying outputs from geodynamic models are dependent on parameters that would be highly variable among samples within real systems (e.g., U and Th concentrations, grain size, $D_{par}$), creating a certain level
of arbitrariness that must be taken into account when comparing model results with data from natural systems. However, we note that the choices of these parameters are no less arbitrary than many of the choices widely adopted for other parameters in geodynamic models, such as the common choice to use an experimentally-derived viscous flow law for quartz aggregates (Gleason and Tullis, 1995) to model the rheology of the entire upper crust.

Other limitations of our software as currently implemented derive from the nature of the geodynamic models used as inputs. Real thermochronometric ages contain additional thermal inheritance that is difficult to capture within such models. In particular, sedimentary rocks frequently contain detrital apatite and zircon grains that reflect the thermal history of the source rocks that were eroded (e.g., Malusà and Fitzgerald, 2020; Rahl et al., 2007), and mimicking this process accurately in a geodynamic model requires not only simulating erosion and deposition but also tracking where eroded material is deposited.
In the example rift inversion model above, such tracking does not take place, and since new sediment has no inherited thermal history, it is simply assigned the thermal history of the material upon which it is deposited via interpolation (Figure 6). Enabling such tracking within ASPECT and/or other geodynamic modeling codes would enhance the ability of our software to simulate the detrital component of thermochronology.

Finally, although designing GDTchron as a Python package has the advantage of making it open-source, readable, and flexible, that comes with a tradeoff in performance relative to compiled programming languages like C++ and Fortran (e.g., Cai et al., 2005). Although we have aimed to optimize GDTchron where possible by vectorizing calculations and using sparse matrices, additional performance improvements could be implemented in future versions. Additionally, the current parallelized workflow using Joblib currently limits calculations to a single node on an HPC cluster, and larger parallel
computations across multiple nodes could be achieved using tools like Dask (Rocklin, 2015) or mpi4py (Dalcin and Fang, 2021). Such improvements would enable GDTchron to be used more efficiently for large high-resolution and/or 3D geodynamic models.

By providing an open-source version of our code on GitHub that is carefully annotated and paired with interactive examples
in Jupyter Notebooks, we hope to encourage community development that will address these and other limitations of the existing software. Recent efforts to gather large compilations of thermochronologic data (e.g., Boone et al., 2023; Lanari et al., 2023) provide ample opportunity for geodynamic modelers to compare their results with observations. Our intention is for this software to be a tool that can grow as needed to fit the needs of both the geodynamic and thermochronology communities for research and training purposes. Our GitHub repository contains instructions for how to use the code as well
as how to make contributions, and we welcome new collaborations for future versions.



## 6 Conclusions

We present GDTchron: an open-source Python package designed to forward model large numbers of thermochronometric ages from the time-temperature (t-T) paths output by geodynamic numerical models. GDTchron adopts previously used modeling approaches and kinetic parameters to estimate apatite (U-Th)/He (AHe), apatite fission track (AFT), and zircon (U-Th)/He (ZHe) ages in a parallelized workflow that can take advantage of multi-core processors and high-performance computing (HPC) nodes. The package can extract t-T paths for unique particles from commonly used output file formats in geodynamic modeling and estimate ages for all parts of the model domain throughout the model run. An interpolation scheme using $k$-d trees allows particles created during the model run to inherit the thermal histories and ages of their nearest neighbors. A simple model of uplift and a more complex model of rift-inversion orogenesis illustrate potential use-cases for this package. The code is designed to facilitate development by the community as needed for improved functionality.

## Author Contributions

DV and PS wrote the Python code. DV, PS, JN, and SB designed and ran the geodynamic models in ASPECT. DV wrote the initial draft, and all authors contributed to revision and preparation of subsequent drafts.

## Code/Data Availability

The source code for the Python package, Jupyter Notebooks used for analysis and figures, and ASPECT parameter files for the geodynamic models are all available in a GitHub repository (https://github.com/dyvasey/gdtchron), which is also archived at Zenodo (Vasey and Scully, 2025). The files in this repository allow the uplift model to be reproduced in its entirety on a personal computer, and the rift inversion model could also be replicated with an ASPECT installation on an HPC cluster. The Jupyter Notebooks include videos showing the full evolution of the uplift and rift inversion models. GDTchron can be installed in a Python environment from the GitHub repository or from the package repositories PyPI and conda-forge.

## Competing Interests

The authors declare that they have no conflict of interest.

## Acknowledgements

Financial support for this study was provided by Tufts University, with scaling tests and ASPECT model runs conducted on the Tufts High Performance Compute (HPC) Cluster (https://it.tufts.edu/high-performance-computing). We thank the





Computational Infrastructure for Geodynamics (https://geodynamics.org), which is funded by the National Science Foundation under awards EAR-0949446, EAR-1550901, and EAR-2149126, for supporting the development of ASPECT. We thank Anne Glerum and Frank Zwaan for their contributions to the rift inversion model used as an example application of GDTchron. We also thank Jim McClain for early conversations with D. Vasey that contributed to the idea for GDTchron.

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
