# Peer review of "Technical note: Geodynamic Thermochronology (GDTchron) - A Python package to calculate low-temperature thermochronometric ages from geodynamic numerical models"

_EGUsphere, 2025_

## Author Comment (AC1)

We thank Chelsea Mackaman-Lofland for the thoughtful, supportive, and constructive comments on our manuscript. In this document, reviewer comments are replicated in black, and our replies are in blue. Following the request from *GChron*, at this stage we indicate only the changes we intend to make in the revised manuscript, which will be submitted soon.

**General comments**

The authors present a new, open-source Python package designed to forward model large numbers of low-temperature thermochronometric ages from time-temperature paths generated by geodynamic numerical models. The manuscript, and Python package, comprise timely and potentially substantial contributions to the fields of geochronology and geodynamics – as the authors mention in the paper, advances in geodynamic numerical modeling have enabled increasingly high-resolution simulations of geological and thermal events across various tectonic settings and thermochronology data provide quantitative constraints against which such models can be validated. The GDTchron Python package presented here expands on thermokinematic modeling approaches (which predict time-temperature histories and low-temperature thermochronology ages based on deformation kinematics and topographic evolution but do not directly incorporate dynamics; Braun, 2003; Braun et al., 2012; Almendral et al., 2015) and previous strategies for integrating time-temperature information from geodynamic models with thermochronology data (which has involved exporting t-T paths from geodynamic models and importing them into thermal history modeling software like HeFTy or QTQt; Ketcham, 2005; Gallagher, 2012) in that it allows for extraction of time-temperature histories, and thermochronology age predictions, for a large number of particles from multiple steps in a geodynamic model.

The paper is well written, the accompanying Python notebooks are well documented/commented and readily accessible via Github, and the figures are generally illustrative.

We warmly thank the reviewer for the supportive comments!

However, this research has several areas that I consider in need of major improvements before manuscript publication and widespread implementation of the Python package by users in the geodynamics and thermochronology communities:

**Specific comments**
1. The evolution of topography can profoundly impact thermal histories and predicted thermochronology cooling ages, particularly for the low-temperature systems that this contribution focuses on (e.g., Braun, 2005; Gilmore et al., 2018). The authors briefly mention this and caution users to implement geodynamic models that account for erosion and sedimentation – and even make changes to the parameters used to approximate topographic evolution to "provide more realistic surface processes" in their rift-inversion orogenesis example model. However, particularly considering potential readers/GDTchron users who may have specific expertise in either geodynamic modeling *or* thermochronology, the authors need to include more detailed information on i) how surface processes and other parameters that affect thermal evolution, like heat production, are incorporated into geodynamic numerical models, ii) the extent to which those choices

may impact t-T histories and predicted cooling ages, and iii) related recommendations or considerations when integrating geodynamic models and thermochronology datasets.

In the revised manuscript, we will be happy to include more background on the implementation of surface processes and heat production in geodynamic models, their impact on cooling ages, and considerations about comparing models with thermochronology data. We provide some initial answers below:

i) Until recently, most geodynamic models approximated surface processes using some version of hillslope diffusion, in which material at higher elevations is redistributed to lower elevations as a basic approximation of erosion/sedimentation. In more recent models, the geodynamic model is coupled with an external surface processes model (such as Fastscape), which may include the stream power law and marine transport/deposition in addition to hillslope diffusion. The inclusion of these latter two processes (especially the stream power law) in a dedicated surface processes model is what we describe as "more realistic."

ii) As our original manuscript and the reviewer allude to, these choices can have a significant impact on cooling ages, since exhumation requires both rock uplift and erosion to bring hotter material at depth to the surface while cooling it. If erosion is too inefficient, rocks with young ages will never be exposed at the surface because they will accumulate too much daughter product as they cool, and if erosion is excessively efficient, ages will be too young because rocks will never accumulate daughter product before being eroded away.

iii) As a result, a major challenge in comparing models with observed ages lies in whether the chosen surface processes model produces patterns of ages that are realistic and whether the patterns or absolute ages are comparable with a given natural system. For example, if given surface process parameters produce reasonable patterns of ages but changing the underlying geodynamic evolution alters those patterns significantly, that could be used to discriminate between competing geodynamic scenarios in a given region. However, unless multiple surface processes implementations have been tested and/or there are observational constraints on the surface history, one could not rule out a surface processes control on the change in pattern.

A more in-depth comparison of the strategies for incorporating surface processes in the Vasey et al. (2024) rift-inversion model, versus the new rift-inversion model showcased in this study, would provide a valuable, and cautionary, case study. Both of these models incorporate the same geodynamic evolution, but produce notably different low-T cooling ages – a change solely related to the mechanism of topographic evolution. These results raise questions regarding the sensitivity of predicted cooling ages, and measured thermochronology data, to geodynamic mechanisms versus surface processes that need to be addressed in this manuscript to help educate potential GDTchron users and establish best practices.

We will gladly outline the differences between the surface processes implementations of these models and their impact on thermochronometric ages in the revised manuscript. In short, the Vasey et al. (2024) model used only hillslope diffusion to smooth topography as a crude approximation of erosion/sedimentation, resulting in accumulation of daughter product outpacing erosion for particles brought to the surface even in areas of significant rock uplift. By contrast, the model we present here is coupled to the landscape evolution model Fastscape and includes erosion following the stream power law, which allows sufficient exhumation to expose material with young ages.

2. Though the authors highlight advances to understanding of thermochronometer systems and diffusion/annealing kinetics in their motivation for this work, they have not incorporated the most widely used and up-to-date kinetics for predicting (U-Th)/He or fission track cooling ages or track length distributions (e.g., Enkelmann and Garver, 2016; Flowers et al., 2009; 2023a; 2023b; Guenthner, 2021; Guenthner et al., 2013, 2014; Ketcham, 2019; Ketcham et al., 2007; Tamer & Ketcham, 2020) in their Python package. This is flagged as a potential future direction, but until these updates are made, users will either be limited in comparisons of geodynamic model predictions to thermochronology data or existing thermal history models, or will need to follow the same workflows of previous studies that exported t-T paths predicted by the geodynamic models for interpretation using other software like QTQt or HeFTy (as performed in Ternois et al., 2021). I encourage the authors to incorporate apatite fission track annealing kinetics after Ketcham et al. (2007), and options to use radiation damage and annealing models for the apatite and zircon (U-Th)/He systems (e.g., Flowers et al., 2009; Guenthner et al., 2013; or more recent updates), in this publication. Baseline parameters for U and Th concentration, grain size, Dpar, and other parameters could draw from standard values– though users would benefit from options to include specific values at specific sample locations to compare with thermochronology data. Routines for the implementation of these kinetic models, and grain parameters, in the PecubeGUI (Bernard et al., 2025) may provide a useful framework for such updates.

Although we certainly see the value in incorporating these updated kinetics into GDTchron, we contend that this initial version of GDTchron is not the best venue for a few reasons:

1. We are aware of a parallel effort within the low-T thermochronology community to create a complete Python repository of diffusion/annealing kinetics. Ideally, this effort would make the routines for age prediction within GDTchron obsolete, in that GDTchron could simply import this package and make use of its updated (and presumably continually updated) kinetics. Creating separate kinetics within the source code of GDTchron thus has the potential to result in significant duplication of effort and could actually impede future integration of these complimentary efforts. We note that while PecubeGUI and other software may include these routines, they are often not open source (although PecubeGUI is written in Python the source code is not currently distributed with the software), and thus it remains non-trivial to incorporate them in other software like GDTchron.

2. For many geodynamic models that are approximating general processes rather than recreating specific systems, the patterns of ages will ultimately be more important than the absolute ages, and the uncertainties inherent in other parameters within the geodynamic models outweigh the relatively small changes in individual ages that will result from the choice of kinetics. As a result, we disagree that results will not be meaningful unless updated parameters are used. Additionally, the improvements in computation time when using simpler kinetics for large numbers of particles may be more important than the inclusion of radiation damage or other more complex kinetics in some models.

3. For the apatite radiation damage accumulation and annealing model (RDAAM) and its zircon counterpart (ZRDAAM) specifically, although these kinetic models do a better job of approximating He behavior in many populations of apatite/zircon (i.e., when there are multiple protracted phases of burial and exhumation and correlations between effective U and age), there may still be situations where the diffusion kinetics employed in GDTchron are appropriate (i.e., when there is monotonic cooling and correlations between grain size and age). Thus, it is not necessarily the case that these models have been entirely superseded by the more recent radiation damage accumulation and annealing models.

We note that GDTchron does allow users to set U, Th, grain size, and mean etch pit diameter (Dpar) for the existing models, although currently only for the entire model domain. We see allowing conditional changes in these parameters throughout the model domain as a logical next step for development.

3. While related references are briefly mentioned and cited (e.g., Ketcham, 2005; 2024; 2025; Gallagher, 2012; Braun, 2003; Braun et al., 2013; etc.), I'd like to encourage the authors to include a more in-depth discussion of how the GDTchron package fits into the framework of related work, including 3-D and 4-D thermokinematic and 2-D (time-depth and time-temperature) thermal history modeling. This may be particularly relevant to discussions and recommendations related to recommendations/best practices for incorporating crustal thermal parameters, surface processes, and disentangling the effects of each system on predicted cooling ages.

We will gladly provide more context about how GDTchron fits within the ecosystem of modeling software for thermochronologic data. In our view, its primary function is to enable quantitative testing of the effects on thermochronometric age of parameters that can be captured in geodynamic but not thermal history or thermokinematic models. These could include far-field driving forces (e.g., extension/compression rate/magnitude, slab pull), rheology (e.g., diffusion vs. dislocation creep, flow laws for different crust/mantle materials), and impact of fluids, melt, and/or phase changes, to name a few examples. This is fundamentally different from thermokinematic models like Pecube in which both motion and topography are generally prescribed. The downside of geodynamic models is that since these exhumation kinematics and surface evolution are not prescribed, it can be significantly more

difficult (though not impossible) to generate models that approximate the surface observations of real systems in detail.

4. Finally, the tendency of the implemented interpolation scheme to assign new syndeformational deposits the same thermal history as adjacent, pre-existing particle paths may be a real limitation in areas that involve significant accumulation of such deposits during polyphase tectonic and geodynamic histories (like the Pyrenees studies cited as examples in this manuscript, Capaldi et al., 2022; Curry et al., 2021). Are there any changes to the interpolation program, as implemented, that would help set up for future changes in ASPECT or other geodynamic models that might include better tracking and ID of particles related to syndeformational sedimentation?

We agree that this is a real limitation, and we will be happy to expand on it in a revised manuscript. We note, however, that this limitation is derived primarily from the geodynamic models themselves rather than from GDTchron. In its current version, GDTchron can track a particle throughout a model run as long as it maintains the same particle ID. In ASPECT, any new particles created (such as those representing deposited sediment) typically receive a new ID, decoupling them from their source. Future implementations of ASPECT-Fastscape coupling would need to have a way to track the provenance of a given sediment particle, either by maintaining the same ID as the source material (in which case no change to GDTchron is necessary) or through another field that connects the particle to its source material. Updating GDTchron to capture this field would be straightforward but would have to be done in response to a corresponding change in ASPECT.

**Technical corrections**

Line 54: Specify that the simplified model involves block uplift here?

We will make this change in the revised manuscript.

Line 55: Because routines incorporating erosion and sedimentation are so important in predicting cooling ages for low-T thermochronometers, I strongly recommend providing more information about what these routines were, & what geological or other evidence supports their use, here in the intro in addition to later in the paper.

We will specify the use of hillslope diffusion, the stream power law, and marine transport/deposition in the revised manuscript.

Line 109 / Section 3.1: What kind of computational resources are required to run GDTchron? Can you provide more information here regarding the types of scenarios that might run on a personal computer/workstation, versus HPC? (I recognize that related information is documented within the Github & other package resources, but would be valuable to include for readers in this section of the paper too)

We will gladly expand on this point in the revised manuscript. The choice of resource will depend on the scale of the model, but in general, we find that determining ages for all particles in the crust (upper ~50 km) at all timesteps in a publishable geodynamic model will likely require use of a high-end desktop workstation (>30 cores) or an HPC cluster node. On the other hand, running a more restricted set of particles or using a shorter or more simplified model may be doable on a more standard personal computer. We see optimization of the code to enable more of the workflow to be possible on a personal computer as an important next phase of development.

Line 141: As a reviewer with expertise in thermal history and structural/thermokinematics modeling but no hands-on experience with geodynamic software like that listed above, I am curious what common inputs and outputs to these software are, particularly as they relate to the temperature field, partical motion, and the development/decay of topography. Even though your target audience may already be familiar with these programs, I recommend including a more comprehensive background/context section that helps frame geodynamic model capabilities, and limitations, for a broader audience. E.g., how is topography typically handled in these systems, and what are recommendations/considerations for evaluating incorporation of topographic change and sedimentation in geodynamic models? In addition, there may be readers with expertise in thermochronology, but not ASPECT etc. modeling, who would like to use GDTchron to extract t-T paths and predict cooling ages based on published geodynamic models to help inform their sampling strategies. Providing a little more context for the geodynamic modeling component would help clarify the intended workflow and capabilities of the GDTchron package.

We appreciate this perspective and will gladly expand on the sorts of parameters that go into these geodynamic models in the revised manuscript. We outline much of the topography and surface processes components in the responses to the "specific comments" above. In general, we note that these models often contain hundreds of input parameters and are highly flexible, which is both their strength and weakness. Such parameters include a wide range of initial and boundary conditions, material properties, and process parameters such as the size and resolution of the model domain, a material model describing how the model deforms, velocity boundary conditions, and temperature models that may incorporate processes like radiogenic or shear heating. The strength of this approach is the ability to have a physical model that is quasi-realistic in which you can isolate any one of these given parameters to see its impact on the final result. The weakness is that many of the choices for these parameters are often semi-arbitrary or poorly constrained, making it potentially difficult to deconvolve their effects on the model outcome.

Line 169: Choose a different word than "flow" when discussing how material exits the top of the box? "Flow" has implications for rheology/deformation style that do not necessarily apply here.

This is a good point, and we will change this to "exit" to avoid potential confusion.

Line 170: By "temperature structure of the box remains constant," do you mean that there is no advection or change in isotherms due to particle uplift & exhumation? It appears so, from the model diagrams? If not, please consider revising this simple example to allow for isotherm advection, which will produce a more geologically accurate case study?

This is correct; this highly simplified model has a static temperature structure. In line with the reviewer's comment above about community familiarity with geodynamic models, we made the choice in this model to substantially reduce the number and complexity of parameters both to make it more easily comprehensible for those who do not do geodynamic modeling and to isolate the effects of rock uplift and mineral/isotopic system on age. We recognize that this is not a geologically accurate model in many respects and will emphasize this more in the revised manuscript.

Line 170: Specify "particle / rock uplift" or "exhumation" here, as there is no surface/topographic uplift in this scenario that maintains flat topography (e.g., has complete erosional efficiency).

We will change this to "rock uplift" in the revised manuscript.

Lines 176-177: Language related to Fig. 5 here is a little confusing – could be interpreted to mean that the timesteps were generated with GDTchron, but they were defined in ASPECT, right? Then the outputs of that model were processed in GDTchron?

This is correct – we will revise this sentence to make it clear that GDTchron is postprocessing the outputs of the ASPECT model.

Lines 179-180, sentence starting with "Ages young with depth…": Phrasing makes it seem like 0 Ma is reached spatially below the PRZ/PAZs, but the unit is time, not space. Change to "Ages young with depth through the partial retention/annealing zones of each thermochronometric system. At the base of these zones, cooling ages are 0 Ma because diffusion and annealing outpace He and fission track production" or similar?

This is a very good suggestion, and we will revise this sentence along these lines.

Line 190 / Fig. 5: I strongly recommend devising a way to show the movement/change in particle positions in this figure - as vectors, a box with no fill showing before/after positions, positions of partial annealing/retention zones before and after uplift, etc. For Y-axis for Figs. 4 and 5, I strongly recommend flipping the axis such that 0 km is at the surface, and values increase with depth (or negative elevation). This recommendation uses the same perspective as structural cross sections and other geodynamic models, and I think would make both thermal structure and predicted cooling age plots more intuitive for readers and potential GDTchron package users.

We will invert the Y-axes in these figures as suggested so that the values reflect depth. We will also add horizontal lines to each plot showing the location of the model surface at the final timestep, which will illustrate when/how much rock uplift is taking place.

Lines 194-196, sentence starting with "One could explore the effects…": Predicted cooling ages for all of these scenarios also depend on the mechanism for incorporating or approximating topographic evolution – which may have just as much impact on the distribution of predicted cooling ages as the aforementioned factors, especially for low-T thermochron data in regions of

relatively slow rock uplift/exhumation (Gilmore et al., 2018; Ketcham, 2025; & references therein).

We agree and will make this point clear both here and in some of the earlier parts of the text where the reviewer suggests emphasizing the importance of the choice of surface processes models.

Lines 200-204: Completely agreed – though I think it would be helpful to articulate how these factors are incorporated into geodynamic models to help educate readers and ensure potential GDTchron users make effective choices when coupling geodynamic models and interpretations of thermochronology data.

We will add to the subsequent paragraph some details of how these features are incorporated into the more complex rift-inversion model. In general, there is considerable variation in how these factors are used across geodynamic models, so we will not be able to describe all possible permutations.

Lines 214-215, sentence starting with "Although thermokinematic models…": Consider including a brief discussion of the similarities & differences of each approach, e.g., Pecube versus GDTchron? What is each good at (e.g., Pecube incorporates a sophisticated framework for estimating topographic evolution, Pecube-D tracks thermal histories of particles involving large amounts of horizontal shortening/extension and complex faulting, statement on how these relate to the conditions of most geodynamic models?)

As described in the response to "Specific Comment" 3, we will be happy to elaborate on the differences between geodynamic and thermokinematic models. As suggested, Pecube does an excellent job of handling topographic evolution and complex prescribed fault systems, whereas geodynamic models can test the effects of a broader range of parameters but in a more generalized way.

Line 220 / Fig. 6: Please clarify that ages are relative to the start time/duration of the model. Consider flipping Y axes for both the model grid (so that 0 km is at Earth's surface and depth/negative elevation increase downwards) and the Surface Ages plots (so that 0 Ma is at the bottom, and older ages are on top - this is the framework commonly used for Pecube-D and other thermokinematic models). What do you mean by "Maximum Age (Ma)" as Y-axis label on bottom charts? Seems like these are just predicted cooling ages?

We will invert the Y-axes of the requested figures in the revised manuscript. We will relabel the Y-axis on the bottom row to be "Age (Ma)" to avoid confusion – the "Maximum Age (Ma)" was included because we use the greatest age at each X-value in the model results as a proxy for the surface age.

Please consider adding some vertical exaggeration to the middle plots, and perhaps intermediate ticks/values to the Plastic Strain and AHe Age gradient keys – predicted cooling ages only use a third of the space, and it is challenging to read these gradient values without zooming in 200+% into the figure.

We will add additional ticks to the colormaps for strain and AHe age. Although we understand that vertical exaggeration would make the middle plots easier to read, it would also distort attempts to compare ages with the locations and geometries of major faults/shear zones (indicated by plastic strain). Our goal with the bottom plots is to make the gradients that may be challenging to see at the surface in the middle plots more readable.

Line 229, sentence starting with "This 2D model…": Which model? That in Vasey et al. (2024) using hillslope diffusion to approximate surface processes, or the revised model in this study that uses Fastscape? Here, it would be valuable to use this example, and the changes made to "provide more realistic surface processes", as a framework for discussing what changes were made to the mechanisms of approximating topographic evolution and their influence on predicted cooling ages.

The parameters described in this sentence apply to both the Vasey et al. (2024) model and the Fastscape-coupled model presented here. We will revise to make this explicit and provide more detail on the implementation of surface processes.

Line 231, sentence starting with "At 16 Myr…": Time reference is unclear here - do you mean  after 16 Myr into the model run, or at 16 Ma? At the end of the sentence, it would be helpful to add "… with a total model run time of 36 Myr" or similar.

In general, we are explicit about using Myr to refer to model runtime and Ma to refer to ages. As suggested by Reviewer 2, we will change all figures and text to use Ma to avoid confusion.

Lines 236-237, "resulting in slightly younger AHe (~10 Ma) and AFT (~14 Ma) ages…": Here, can also emphasize that these predicted ages are partially reset with respect to inherited model age?

This is a good point, and we will make this point explicitly in the revised manuscript.

Line 240: Update "…lithospheric thickening and uplift" to "lithospheric thickening, uplift, and erosion"

We will make this change in the revised manuscript.

Lines 259-261, "Implementing these alternative kinetic models…": Why not incorporate the most widely used kinetic models in this first release, especially if kinetic models are styled after those used in Ketcham (2005)? Doing so would significantly expand the potential for comparisons between predictions based on geodynamic models and thermochronology data for specific study areas.

We agree that including these kinetic models would be useful, though we do not think this initial release is the best venue to do so. See our response to "Specific Comment" 2 for details.

Lines 263-265, "Importantly, the ages produced when applying outputs from geodynamic models are dependent on parameters that would be highly variable among samples within real systems (e.g., U and Th concentrations, grain size, Dpar)…": Here, viable solutions might include incorporating values based on best practices (e.g., for grain sizes for He analyses) or values based on standards, then allowing for users to adjust GDTchron sample parameters to best match thermochronology data of interest? This suggestion is inspired by similar options in thermokinematic modeling programs including PecubeGUI (Bernard et al., 2025).

Currently, GDTchron does allow the user to modify U and Th concentrations, grain size, and Dpar from default values for a particular model run, and we will make that explicit in the revised text.

**References not included in the manuscript:**

Almendral, A., Robles, W., Parra, M., Mora, A., Ketcham, R. A., & Raghib, M. (2015). FetKin: Coupling kinematic restorations and temperature to predict thrusting, exhumation histories, and thermochronometric ages. AAPG Bulletin, 99(8), 1557-1573.

Bernard, M., van der Beek, P., Colleps, C., & Amalberti, J. (2022, May). PecubeGUI: a new graphical user interface for Pecube, introduction and sample-specific predictions of apatite (U-Th)/He and 4He/3He data in the Rhone valley, Switzerland. In EGU General Assembly Conference Abstracts (pp. EGU22-2277)., 524–525, 1–28, https://doi.org/10.1016/j.tecto.2011.12.035, 2012.

Gilmore, M. E., McQuarrie, N., Eizenhöfer, P. R., & Ehlers, T. A. (2018). Testing the effects of topography, geometry, and kinematics on modeled thermochronometer cooling ages in the eastern Bhutan Himalaya. Solid Earth, 9(3), 599-627.

Ketcham, R. A. (2025). Incorporating topographic deflection effects into thermal history modelling. Geochronology, 7(3), 449-458.

---

## Author Comment (AC2)

We thank Kendra Murray for the thoughtful, supportive, and constructive comments on our manuscript. In this document, reviewer comments are replicated in black, and our replies are in blue. Following the request from *GChron*, at this stage we indicate only the changes we intend to make in the revised manuscript, which will be submitted soon.

This manuscript presents a new open-source Python package for predicting cooling ages for AHe, ZHe, and AFT chronometers from time-temperature (tT) histories extracted from the results of geodynamic numerical simulations. It particular, it addresses the challenge of the large number of tT paths that are produced by such simulations that need to be forward modeled in order to validate geodynamic simulations with thermochronologic observations. Using established approaches for numerically integrating diffusion and annealing behaviors along a particle tT path, this modeling package can take in outputs from common geodynamic modeling software. It also addresses some of the known problems with extracting specific particle tT paths from such numerical results using an interpolation scheme. This contribution explicitly presents an open-source starting point for developing tools that are capable of jointly leveraging the power of geodynamic simulations and thermal history analysis of low-temperature thermochronologic data. As such, some features (such as the implementation of thermochronometer kinetics models) are overly simplified, with the invitation for community-generated innovation and updates that can implement more complex and realistic behaviors as needed.

The central challenge that this new Python package is designed to address is interesting, important, and relevant within the scope of GChron and the Special Issue it was submitted to. The accompanying online documentation and demonstrations are a laudable companion resource. Creating accessible tools to bridge the divides between the geodynamic and thermochronologic communities has the potential to support advances in multiple geoscience disciplines. However, as a thermochronologist, there are a few things about the design and description of the proof-of-concept examples presented here that I think would benefit from some adjustment, which I describe below. I think with revision, this will be an excellent contribution.

We warmly thank the reviewer for their support of our contribution for inclusion in this special issue for *GChron*.

**MAJOR COMMENTS**

1. Before walking through specific examples, I have one general comment for all figures and corresponding text: the axes on many graphs are presented in the opposite sense from what is typical for thermochronological data and exhumation studies, which overcomplicates the presentation of the numerical results. In most cases, this is because the depth (y position) and time (elapsed model time) information is being directly exported from the simulations without being converted to and presented in a geological reference frame. These conversions are simple, but for thermochronological data, this matters! It matters because depth below the surface corresponds to closure behavior, and cooling ages are in Ma (millions of years before present) not model time (Myr elapsed). Users (and readers) need to be able to efficiently evaluate if the numerical results "make sense" thermochronologically. I provide some details below, but overall I suggest that

unless it is essential to present the modeling reference frame, all the results for each example be reframed as depth = depth below the surface and time = geological time (Ma). I realize that following these suggestions may require parallel updates to the very nice online documentation the authors have prepared; however, I also anticipate that just a few simple lines of code are required to flip axes or convert model depth to geological depth.

We will be happy to revise all figures that employ Y position so that they indicate depth below the surface. We will also reframe all references to model runtime in Myr to be geologic ages in Ma.

**2. Forward modeling example (Section 2.3)**

1. The forward modeling demonstration has a strange starting condition from a thermochronologic perspective that is not explained. The modelled tT path starts at 100 C, but it is used to predict ages for thermochronometers that are completely (ZHe), partially (AFT), or not at all (AHe) closed at 100 C. The manuscript should either: be explicit about the assumptions of such a model design and describe clearly how the consequences of these assumptions produce the model results; or, have a starting condition that is hotter than the temperature sensitivity of all systems being modelled. This is not sufficiently discussed in the current text. For example, at line 95, the text mischaracterizes the reason why the ZHe age is predicted to be 34.8 Ma. "For the zircon (U-Th)/He (ZHe) system, this path results in an age of 34.8 Ma, corresponding to nearly the full duration of the 35 Myr history, given that the sample remains at a temperature *largely below the ZHe* partial retention zone throughout this history." (emphasis added). The starting condition (and isothermal hold T) is not "largely" colder than the ZHe partial retention temperatures for the Reiners et al. 2004, it is much much colder (at least 50C colder) than even the coldest part of the PRZ for these kinetics (and for the current ZRDAAM kinetics too, for such as tT history). From a thermochronological perspective, this zircon grain simply "appears" at 100C at 35 million years ago, and accumulates He that whole time. Having the model start in the midst of the AFT PRZ is even more complicated and potentially problematic from my perspective. Unless this is intentional? If so, why?

This choice of example model was intentional because of the behavior the reviewer describes, and we will be sure to revise the manuscript to make the rationale for this choice more explicit.

We chose this example because it produces ages in each system that intuitively make sense given the variations in their partial retention zones (PRZ) or partial annealing zones (PAZ). As the reviewer states, ZHe is completely closed at 100°C and accumulates He throughout entire the modeled t-T path, AFT is partially closed and so has a complicated age that does not correspond to a particular thermal event, and AHe is not at all closed and so has an age that corresponds to the prescribed cooling event. As a result, all 3 systems have distinctly different

ages, and the explanations for those distinctly different ages are intuitive given what is known about each system's kinetics.

We will remove the word "largely" from the discussion of the ZHe PRZ.

2. When I put this tT history into HeFTy (v 2.3.1), using the Ketcham 1999 annealing model, a Dpar of 1.75, and c-axis projection (and Ketcham 2003 c-axis projection) it predicts an AFT age of 19.8 Ma, not 21.3 Ma. (The predicted tracklength distribution (Fig. 1c) is not reported in numbers so I cannot compare that to HeFTy's prediction.) Section 2.2 describes that Ketcham's 2005 approach (i.e., HeFTy) is being used for the AFT system here, but was this new code benchmarked against HeFTy? If I'm not setting up the AFT correctly in HeFTy to mirror what is being done here, then more information is needed. Such a difference in ages may not be geologically important, but from a numerical calculation perspective it is essential to know why this discrepancy arises. Starting the history from within the AFT PRZ makes this discrepancy additionally difficult to diagnose.

GDTchron is benchmarked against HeFTy, using the example t-T paths that are provided by Ketcham (2005), and there is very good agreement between GDTchron and HeFTy for these examples. In fact, these examples are currently used as unit tests within GDTchron, such that no changes to the source code can be implemented if they result in ages that do not match the HeFTy ages. We will revise the manuscript to indicate this benchmarking.

We agree that there is a small discrepancy between HeFTy and GDTchron for this particular AFT age in our example t-T path, and that the history beginning within the AFT PAZ likely plays a role. It is difficult to diagnose the exact numerical reason for the discrepancy because HeFTy is not open-source software, and so we cannot directly compare our routines with its source code. One possibility is that the time intervals we input into GDTchron for this this example do not precisely align with the time intervals HeFTy uses for its calculations of annealing behavior. As described in our response to Reviewer 1, we are aware of a separate community effort to create a Python repository of thermochronology kinetics that we think would make it easier to assess the causes of such discrepancies.

**3. Simple "uplift"**

1. This scenario is repeatedly described as "uplift," but thermochronologic ages never directly document uplift. They only document rock cooling, which in this scenario is driven by rock exhumation that happens when the imposed *rock* uplift occurs without any corresponding *surface* uplift (in other words, rock exhumation). Thus, this is a "simple exhumation" scenario, and should be described as such, starting in the abstract. I suggest never using the word "uplift" without specifying "surface" or "rock", to avoid this confusion. See England and Molnar, 1990 (Surface uplift, uplift of rocks, and exhumation of rocks: Geology, v. 18, p. 1173–1177)

This is a good point, and we will be sure to revise the manuscript to be very explicit about when we mean surface uplift and when we mean rock uplift. The reviewer is correct that this model is better described as "simple exhumation" since there is rock uplift without surface uplift, and we will revise the text to reflect this throughout.

2. The model design description at lines 168-170 states: "At 10 Myr, the bottom of the box is pushed upwards at a rate of 1 mm/yr, allowing material to flow out the top of the box while the temperature structure of the box remains constant," If I understand correctly, in the geodynamic model used to generate the tT paths modeled here, a simple linear geothermal gradient was held constant despite a massive transient change in rock exhumation rate, from 0 km/Myr to 1 km/Myr and then back to 0 km/Myr. By imposing rock uplift and exhumation without any corresponding change in the geothermal gradient, this example is an extraordinarily unrealistic departure from how we know the Earth works. Especially considering the imposed rapid exhumation rate: the Peclet number for this scenario [Pe = (exhumation rate \* lithosphere thickness)/thermal diffusivity, which describes the competition between heat advection and conduction] is close to 1 (assuming a standard thermal diffusivity of 25 km2/Myr), and so this is nearly an advection-dominated system that would not even be well represented by using a steady-state solution of the conductive geotherm for a 1 km/Myr exhumation rate. These considerations and their consequences for cooling age patterns have been quantified and evaluated by the thermochronologic community for decades (see for example the "Quantitative Thermochronology" book by Braun et al published in 2006). Given (my admittedly under-informed sense of) the abilities of geodynamic models, I'm surprised at this oversimplification, and surprised that the thermal limitations of this geodynamic result (and the corresponding age predictions) are not acknowledged. The main point of this manuscript is to demonstrate how the new tool addresses some of the major challenges of transferring particle-path tT predictions from geodynamical tools to thermal history analysis tools. And there will necessarily be some simplifications. However, given that the audience of this Technical Note spans both thermochronolgists and geodynamicists, I think it is essential that the examples use simplified, but still physically reasonable, thermokinematic scenarios.

We agree that this is an unrealistic thermal regime for the reasons the reviewer describes and will revise the manuscript to make that more explicit. We note that this example model is not intended to demonstrate the full capabilities of geodynamic models, which typically do employ much more complex and realistic temperature models, but rather to illustrate how the results of a geodynamic model, regardless of complexity, can be output to GDTchron to predict ages. In this case, a model with a static linear geothermal gradient is useful precisely because the expected thermochronometric ages are easy to predict and understand for those who are not experts in thermochronology (which is also a component of our target audience). Such users may not have thought about Peclet numbers, but

they can understand that if you know roughly at what depth particular isotherms occur, then you can connect cooling ages to advection of material from particular depths.

We note that the subsequent rift-inversion model is a much more complex geodynamic model with a more realistic temperature model like the one the reviewer describes. We will make this more explicit in the revised manuscript.

- 3. Is 10 Myr sufficient for these thermochronometers to achieve equilibrium at partial retention/annealing temperatures for each system? In other words, how arbitrary is this choice of duration, and does that matter for this demonstration?
  - 10 Myr is a fairly arbitrary choice that is mainly used so that there is an opportunity for He/FT to accumulate both above and within the PAZ/PRZ prior to the rapid "exhumation" event. We will make this rationale explicit in the revised manuscript.
- 4. The text should be more clear about reporting 'model time' (Myr elapsed since the start of a simulation) vs. 'geological time' (Ma, what the cooling ages are documenting and how we think about geodynamic histories); see also Major Comment #1 above. The use of Ma vs Myr is consistent and correct, but the practical difference between these concepts is not explained. For example, Figure 5 is mixing these two opposite concepts. My suggestion: Unless it is essential for the reader to see the 'model time' framing of the design of these numerical simulations, I suggest the authors simply convert everything into 'geological time' for us. This matters because thermochronologists always start examining data (whether synthetic or real) by comparing cooling ages to intervals of geological time that are of interest. In this simple example, I'm looking to see how the predicted cooling ages at the end of the model run ("20 Myr – After Quiescence" in Fig. 5) document (or not) the imposed exhumation event in order to assess the rigor and utility of this approach. But, rather than just looking at Figure 5, I have to do the extra work of converting the geodynamic model time to geological time, or vice versa. (Geologic time = How long ago did the exhumation event start and end?) I also have to know, as a reader, that this conversion is necessary.

We will revise the manuscript to report model runtimes as geological times in Ma to reduce the potential for confusion.

- 4. Complex model, Figure 6
  - 1. For the "AHe Ages" plots, why is the result shown for 150 km thickness? It seems like He ages are only calculated for the top 50 km, and even that is an order of magnitude greater lithospheric thickness than these ages are sensitive to. The relatively near-surface details (between the surface and the effective closure of each system) is what is important for assessing the utility of this tool, but that cannot be seen here.

These plots are shown at the same scale and covering the same area of the model domain as the top row, so that it is easy to directly compare other outputs of the model (such as location of shear zones/faults) with the distribution of thermochronometric ages. Because that makes it difficult to see the patterns at the surface, we include the bottom row of plots to show the surface ages across the model domain.

He ages are only calculated for the top 50 km because, as the reviewer states, anything deeper would have no change of every actually accumulating He. We use 50 km to conservatively ensure that we capture any material that might have entered the PRZ sometime during the 36 Myr model runtime.

2. For the "Surface Ages" plots, why is age increasing down the y-axis? This is the opposite of how such data are usually visualized, including in similar tools (see for example, Fig. 6 from this paper by McQuarrie et al: <a href="https://agupubs.onlinelibrary.wiley.com/doi/full/10.1029/2018TC005340">https://agupubs.onlinelibrary.wiley.com/doi/full/10.1029/2018TC005340</a>)

We will revise these plots so that age is decreasing down the Y axis.

- 5. Description and simplification of the thermochronometer kinetics models
  - 1. Sections 2.1 and 2.2 would benefit from reporting the nominal closure temperature ranges for each system, given the kinetics being used in this version of GDTchron.

We will revise these sections to include approximate closure temperatures for these systems.

2. Simplifying the diffusion and annealing kinetics models being implemented in this first version of GDTchron makes sense, and I support the approach taken here as a first step (so long as it is not the last). But, I think this choice warrants more discussion up front, in the methods section of the paper, to help convince thermochronologists of the basic utility of the current GDTchron package despite the limited kinetics, highlight its potential for future adaptation, and describe the consequences of simplifying kinetics in this way for those who are not familiar with chronometer kinetics. For He diffusion kinetics, for example, the 2000 AHe kinetics parameters and 2004 ZHe kinetics parameters being used here are extremely out of date; from the perspective of thermal history analysis, these kinetics models have been superseded by the radiation damage accumulation and annealing models and the old "static" kinetics parameters should never be used to interpret real thermochronologic data, especially for scenarios with certain styles of thermal histories. But of course, the choice of kinetics model only ends up being important in some contexts. So, I suggest that the authors move much of the information presented at the beginning of section 5 (lines 250-260) into section 2.1 and 2.2. The authors could support this choice by discussing how other numerical tools (e.g., Pecube, age2exhume) have navigated this challenge, and in

what types of thermal histories / geological scenarios not using the radiation damage kinetics models matters most.

We appreciate the reviewer's acknowledgement of these kinetics as a good first step, and we agree that employing more updated kinetics in future versions will increase the utility of GDTchron. As described in the response to Reviewer 1, we anticipate that a parallel community effort to compile kinetics in a Python repository will provide an opportunity to integrate these kinetics into GDTchron without a large duplication of effort. We will move information from section 5 into section 2 in the revised manuscript to better clarify the consequences of this initial choice of kinetics, as well as the scenarios where these kinetics are still appropriate. We do not necessarily agree that these kinetics should never be used to interpret real data in cases where there is reason to expect monotonic cooling and where there is a strong correlation between grain size and age, for example, but we acknowledge that in many scenarios they are not the preferred choice.

**MINOR COMMENTS**

Figure 1b: Can the user adjust the grain size and composition? Can different kinetics be chosen?

The user can assign grain size and composition, though in the current version they apply only to the entire model domain. We will make that explicit in the revised manuscript. Different kinetics could be chosen once they are implemented in future versions.